# An Efficient Calibration System of Optical Interferometer for Measuring Middle and Upper Atmospheric Wind

Guangyi Zhu [1,2,3], Yajun Zhu [1,2,3,4,*], Martin Kaufmann [5], Tiancai Wang [1,2,3], Weijun Liu [1,2,3,4] and Jiyao Xu [1,2,3,4]

1    State Key Laboratory of Space Weather, National Space Science Center, Chinese Academy of Sciences, Beijing 100190, China
2    Key Laboratory of Solar Activity and Space Weather, National Space Science Center, Chinese Academy of Sciences, Beijing 100190, China
3    University of Chinese Academy of Sciences, Beijing 100190, China
4    Hainan National Field Science Observation and Research Observatory for Space Weather, National Space Science Center, Chinese Academy of Sciences, Beijing 100190, China
5    Institute of Energy and Climate Research (IEK-7), Jülich Research Centre, 52425 Jülich, Germany
*    Correspondence: y.zhu@spaceweather.ac.cn

**Abstract:** Detection of the Doppler shift of airglow radiation in the middle and upper atmosphere is one of the most important methods for remote sensing of the atmospheric wind field. Laboratory and routine field calibration of an optical interferometer for wind measurement is very important. We report a novel calibration system that simulates a frequency shift of airglow emission lines introduced by wind in the middle and upper atmosphere for calibrating passive optical interferometers. The generator avoids the shortcomings of traditional motor-driven Doppler-shift generators in terms of stability and security while improving accuracy and simplifying assemblies. A simulated wind speed can be determined simultaneously using the light-beat method. The wind error simulated by the generator mainly comes from the light source, which is about 0.63 m/s. An experimental demonstration was conducted using a calibrated Fabry–Perot interferometer and showed that the root mean square of the measurement uncertainty is 0.91 m/s. The novel calibration system was applied to calibrate an asymmetric spatial heterodyne spectrometer (ASHS)-type interferometer successfully. The results demonstrate the feasibility of the system.

**Keywords:** middle and upper atmosphere; wind detection; asymmetric spatial heterodyne interferometer; doppler-shift generator; calibration

## 1. Introduction

Wind is a critical parameter for characterizing the dynamics and energetics of the middle and upper atmosphere. Wind data, as the basis of atmospheric models, help to understand the momentum and energy transfer processes of different ionized characteristic particles in the upper atmosphere and provide support for space environment forecasts [1–3]. Atmospheric wind measurement has become widely recognized under the demand for securing satellite communications [4,5].

Currently, the optical passive sounding technique is a common method for regularly obtaining neutral winds in the upper atmosphere [6,7]. The basic principle is to estimate the Doppler shift of an airglow emission line. Different types of interferometers have been developed to detect neutral winds in the middle and upper atmosphere, such as the Fabry–Perot interferometer (FPI), Michelson interferometer (MI), and asymmetric spatial heterodyne spectrometer (ASHS)-type interferometer. These instruments are widely used for space-borne and ground-based wind measurements. The first satellite-based passive optical detection of upper atmospheric wind was achieved by the FPI onboard the DE-2 satellite in 1982 [8]. Subsequently, various types of satellite-based interferometers have been developed to measure winds in the mesosphere and thermosphere, such as the FPI-type

TIDI on TIMED, Michelson-type instrument WINDII on UARS, and ASHS-type instrument MIGHTI on ICON [9–11]. Many ground-based interferometers have been deployed worldwide to observe middle and upper atmospheric winds in mid-latitudes [6,12–14]. The wind precision of interferometers has been improved to m/s level from the initial dozen m/s level with the progress of technology in the past decades.

One of the most important tasks in the development and operation of optical passive wind-sounding instruments is to evaluate their wind-measurement capability and to determine their accuracy, including precision and systematic error [15]. Therefore, a tunable and high-precision calibration system to simulate upper-atmosphere winds is required. A type of motor-driven calibration system to generate Doppler frequency shift is often used for the purpose of wind calibration, consisting of a frequency-stabilized light source and a motor-driven reflective disk in general. The light source is irradiated on the high-speed rotating disk covered with a micro-lens film at a certain angle [16,17]. Different wind speeds can be achieved by adjusting the motor speed. However, the motor-driven calibration system has several flaws, which may result in an increase in uncertainty and problems during the operation. The assembly angle of the reflective disk and the angle of incidence and reflection are difficult to measure accurately, which causes significant errors on the m/s level. A statistical error of 3 m/s was found by using a motor-driven system to calibrate an ASHS-type interferometer [18]. Another study discussed the uncertainty of the rotating disk, including a random error of 0.17 m/s and a system error of 1.36 m/s at a speed of 102 m/s [19]. Furthermore, the measurement process is complicated, including determining the angle of incidence and spot position as well as collimating the incident light and collecting the reflected light. Another problem is that the reflector disk rotates at a high speed, which increases the security risk of the experiment and the instability of the entire system. Last but not least, the phase drift of the motor-driven system caused by rotating instability cannot be monitored during calibration, which introduces an additional error. On the other hand, an acousto-optic modular (AOM) was used to test phase responses of the mini-MIGHTI in the laboratory, and the frequency of a 633 nm He-Ne laser was shifted by 80 MHz, corresponding to a simulated Doppler speed of 50.6 m/s [20]. However, there has not been a complete discussion on the calibration of the system, uncertainty, or effect.

To popularize the new Doppler-shift calibration system, a series of validation experiments were carried out. We presented more details to demonstrate the effect and application value of the calibration system. The concept of the wind simulator is presented in Section 2. In Section 3, the experimental demonstration is described, including its physical basis, experimental setup, data processing, and results of the experimental comparison. Section 4 presents the principle and process of calibrating an interferometer. Finally, an evaluation of the new instrument is provided in Section 5.

## 2. Instrument Concept

The new setup of the Doppler-shift system has been developed in the laboratory to simulate neutral winds in the middle and upper atmosphere, which consists of a frequency-stabilized He-Ne laser, an isolator, an acousto-optic frequency shifter (AOFS), a diffuser, and three lenses. The new setup of the system takes advantage of the AOFS, which can modulate the frequency of lasers in the order of hundreds of MHz, as shown in Figure 1. The AOFS has several advantages over a motor-driven system, such as pure frequency, high efficiency, and ease of operation.

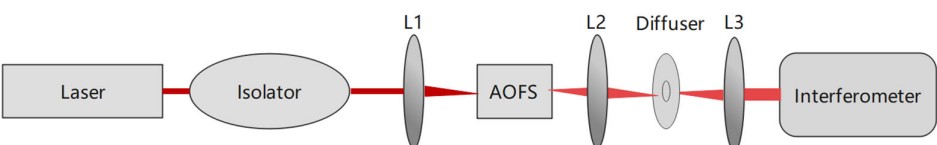

**Figure 1.** New setup using acousto-optic frequency shifter to generate Doppler shifts in the laboratory.

The He-Ne laser used in this setup provides a light source at 632.8 nm with a frequency stability of $\pm 1$ MHz, resulting in a wind uncertainty of about $\pm 0.63$ m/s. The outgoing light first passes through an isolator to prevent reflected light from entering the laser. Then, the laser beam is focused onto the AOFS by the lens L1 to maximize the modulation efficiency of the RF carrier and to reduce the light transit time. The beam should be incident at the Bragg angle to obtain the maximum diffraction efficiency [21]. The frequency range modulated by the AOFS used here is 100–200 MHz, which can simulate wind speeds in a range of 63.28–126.56 m/s [22]. The lens L2 focuses the beam onto the rotating diffuser to suppress laser speckle, and the lens L3 collimates the laser beam for the interferometer. The simulation of the wind at different speeds can be achieved by adjusting the AOFS.

## 3. Experimental Demonstration

According to the working characteristics of the AOFS, it is important to confirm whether the frequency shifts are the same as the values, as given by the controller. The frequency of visible light reaches $10^{14-15}$ Hz, which is significantly higher than the response frequency of all currently available photodetectors. Therefore, we choose an indirect approach to detect the frequency shift modulated by the AOFS, and here, the light-beat method is used to measure the frequency shift.

### 3.1. Experimental Setup

To measure the change in frequency when light passes through the AOFS, a new setup was built, as shown in Figure 2. The laser beam passing through the isolator was divided into two beams using a beam splitter (BS1). One beam passes through the AOFS, and the other one is reflected by two mirrors without passing through the AOFS. Another beam splitter (BS2) mixes the two beams, which are then focused by L3 onto a photodetector integrated circuit (PDIC). The PDIC is a Si-amplified detector that converts optical signals into electrical signals. Finally, the photodetector was connected to an oscilloscope to capture the light-beat signal. The Si-amplified detector used in the experiments had an impulse response time of 1 ns and a bandwidth of 380 MHz. The sampling frequency of the oscilloscope was 1.25 GHz, which allows to detect 100–200 MHz signals. To improve the resolution of the frequency-domain spectrum, it is necessary to ensure a certain sampling time, which was 1.25 ms per sample in the experiment, corresponding to a frequency resolution of 0.8 kHz.

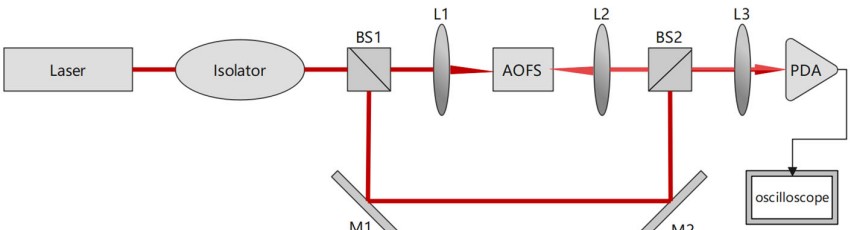

**Figure 2.** Experimental setup for Doppler shift determination by light-beat method.

### 3.2. Beat Frequency Obtaining

The light-beat frequency can be obtained using a fast Fourier transform (FFT) with the amplitude of the signal captured by the oscilloscope. As shown in Figure 3, with a 100 MHz frequency shift, for example, a spectral peak can be easily detected, and the corresponding frequency is shown in the horizontal axis in the figure. Here, the energy of the DC term is much higher than the energy of the target frequency, most likely because the light of the two frequencies does not exactly overlap on the detector, allowing for a greater collection of light from a single frequency.

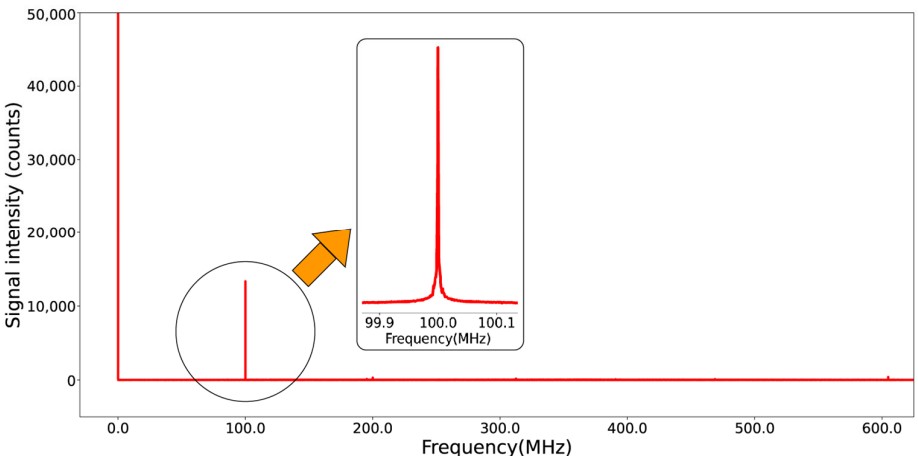

**Figure 3.** Determination of light-beat frequency with FFT (100 MHz as an example).

### 3.3. Experimental Processes and Results

In the first experiment, the light-beat signal was captured using the setup shown in Figure 2. The AOFS operates from 100–200 MHz, corresponding to a wind speed of 63.28–126.56 m/s. The frequency shift was set at an interval of 4 MHz each time, which is shown in the vertical coordinate. According to the method described in Section 3.3, the frequency of the detected light-beat signal is calculated and shown on the horizontal coordinate. The results are presented in Figure 4, where the diamond-shaped points are the measurement points, the dashed line indicates the linear fit of the measurement points, and the solid line shows the deviation of the light-beat frequency from the set frequency shift (note that the unit is kHz). The root mean square of the deviation in the measurements was 1.6 kHz. That means the frequency error of the light source dominates the uncertainty budget of this method rather than the errors introduced by the AOFS.

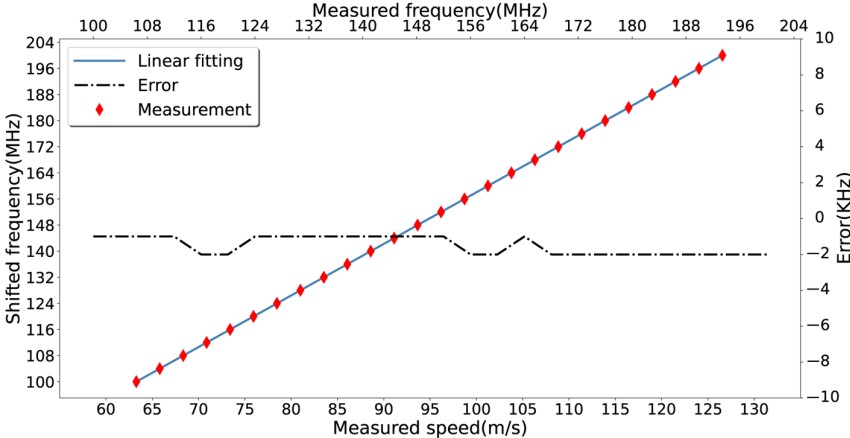

**Figure 4.** Measurement results of the light-beat method.

In order to demonstrate the feasibility of the new Doppler-shift system, another experiment was conducted by using a well-calibrated interferometer for wind measurements in the middle and upper atmosphere, which is installed in Xinglong station near Beijing [23,24]. This experiment setup is given in Figure 1, using the same light source as in the previous setup. Figure 5 shows the FPI measurement results. The root mean square of the deviation between the AOFS and FPI measurements is $\sigma = 1.49$ MHz, corresponding to a wind speed statistical error of 0.91 m/s. The measurement results demonstrate the suitability of the light-beat method for the calibration of instruments obtaining wind speed from Doppler-shift measurements.

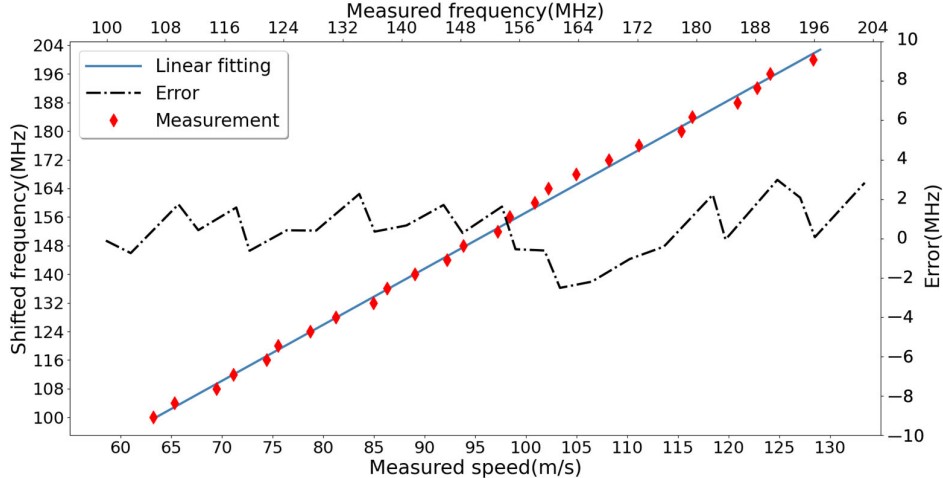

**Figure 5.** Measurement results of the calibrated FPI.

## 4. ASHS-Type Interferometer Calibration

We use the setup presented above to calibrate a newly built ASHS-type interferometer (Figure 6, Wei et al., 2020) [25]. As shown in Figure 6, an approximate plane wave entering the interferometer is divided into two beams by the beam splitter BS1. After going through the fixed prisms for enlarging the field, they are diffracted at the two gratings, respectively. Then the two reflected beams are combined and recorded by the detector as an interferogram that carries the frequency of the light source. The asymmetry of the two interferometer arms allows for the detection of small frequency changes in the input signal as phase shifts in the interferogram. The wavelength shift can be calculated from the phase using the Fourier transformation. Wind speed can be calculated from the phase shift by the following expression [15]:

$$v = \frac{c}{2\pi\sigma_0 L}\Delta\phi, \tag{1}$$

where $c$ is the speed of light, $\sigma_0$ is the wavenumber of the emission line, $L$ is the optical path difference (OPD), and $\Delta\phi$ is the phase difference between the interferograms with and without Doppler shift.

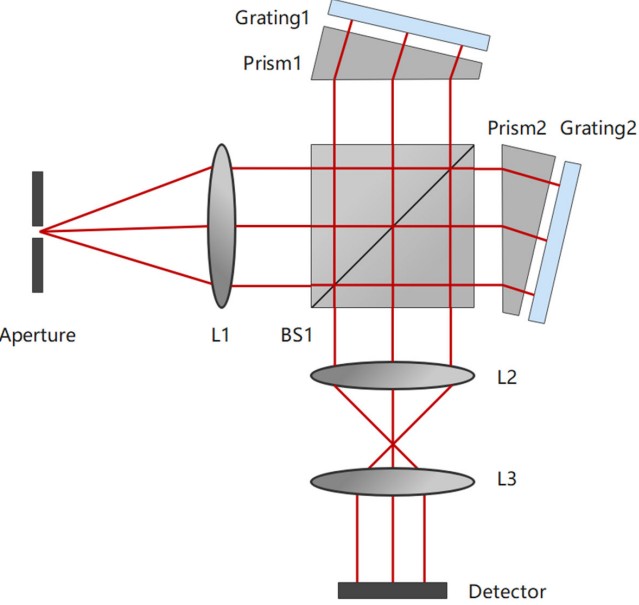

**Figure 6.** Schematic of an ASHS-type interferometer configuration.

The coefficient $\frac{c}{2\pi\sigma_0 L}$ can be considered a factor, being defined as the sensitivity, which decides the accuracy of the inversion. However, the OPD is a fuzzy parameter for some actual reasons. In addition to the tolerances in practical fabrication, errors may also be introduced in the gluing and UV curing processes during the instrument assembly. The prisms and gratings of the interferometer are thermally sensitive, so the OPD will also change under different working conditions. The OPD can be expressed as

$$L = D + 4x \tan\theta \tag{2}$$

where $D$ is an effective optical path difference at the center of the detector considering the optical dispersion effect, $x$ is the horizontal position on the detector, and $\theta$ is the Littrow angle of the instrument, which may also change after the assembly. The designed effective optical path difference $D$ of the assembled interferometer we used is about 32.4 mm at the path center, which will change depending on the thermal performance and imaging properties, as well as the Littrow angle $\theta$ [25]. According to our simulation, a 0.1 mm difference in OPD corresponds to an inverting speed error of about 0.51 m/s based on Equation (1) and the parameters of the system. Therefore, a fine calibration is essential for the newly developed ASHS-type interferometer.

To calibrate the ASHS instrument, light from a neon (Ne) calibration lamp is superimposed on the signal passing through the AOFS (Figure 7) to detect and to correct for thermal drifts in the ASHS [26]. A narrow bandpass filter is used to isolate the target emission line at 630.48 nm and to suppress other emission lines.

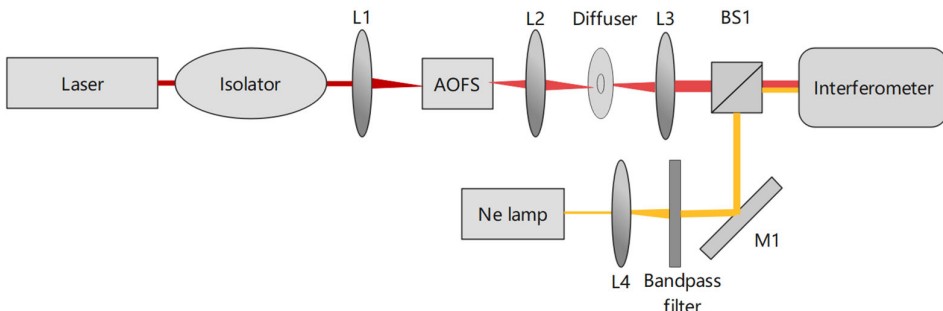

**Figure 7.** Measuring and calibrating setup for an ASHS-type interferometer.

In order to calibrate the interferometer, we measured ten groups of data, and each group includes 26 interferograms at different speeds in the range of 63.28 m/s to 126.56 m/s. The sensitivity of the interferometer in terms of Doppler speed per phase shift was calculated using the designed nominal OPD and the Littrow angle, which is $\frac{c}{2\pi\sigma_0 L} = 845.82\,\frac{m/s}{rad}$. We derived the Doppler speed from one group of the data using the calculated sensitivity, as shown in Figure 8. A positive bias was found. The maximum error is 1.74 m/s, and the root mean square of the deviation is 1.01 m/s. This means that the nominal factor used for retrieval is negatively biased.

According to Equation (1), we used the remaining nine groups of data to fit a more precise factor of the coefficient $\frac{c}{2\pi\sigma_0 L}$. Figure 9 shows the average of measurements and their linear fitting result, where an expectation line is also given for comparison and the slope of it represents the designed sensitivity. Please note that each data point in the figure is an average value of nine retrieved Doppler speeds. The calibrated factor changes to $853.12\,\frac{m/s}{rad}$.

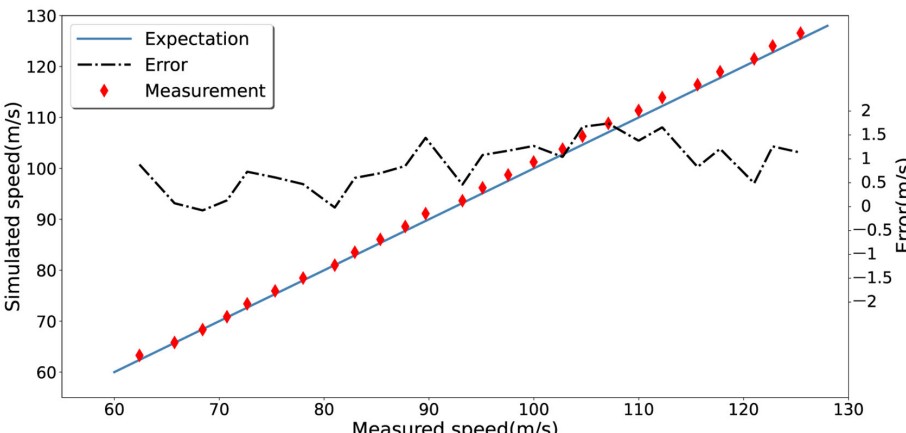

**Figure 8.** The simulated and measured Doppler speed with an ASHS-type interferometer before calibration.

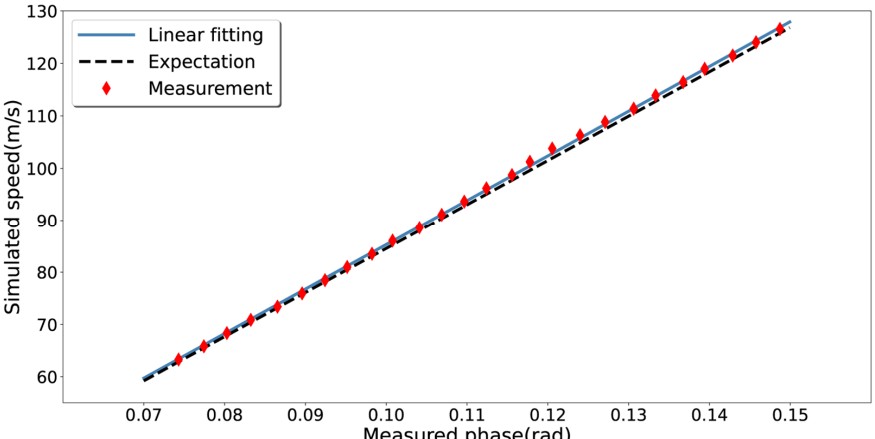

**Figure 9.** The mean of the measurements and the comparison of the sensitivity.

After implementing the calibration, we derived the Doppler speed from the same data used in Figure 8 with a sensitivity of $853.12\frac{\text{m/s}}{\text{rad}}$, which is given in Figure 10. Compared with the previous inversion in Figure 8, clearly, the errors were reduced to less than 1 m/s. The root mean square of the deviation was 0.43 m/s, and no systematic error was found.

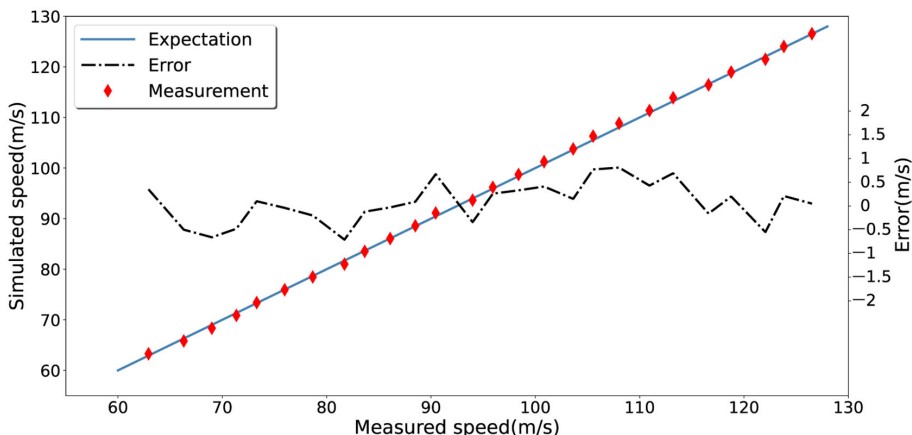

**Figure 10.** The simulated and measured Doppler speed after calibrating using the system.

The error of the instrument reduces from 1.01 m/s to 0.43 m/s for the same set of data. The result of the calibration demonstrates the feasibility of the novel Doppler-shift system,

and a routine calibration with this system is requisite for interferometers, especially for those running long periods in field stations.

## 5. Conclusions

We present a new Doppler-shift calibration system concept for optical passive wind detection interferometers. To simulate the upper atmospheric winds, Doppler-shift generation was provided by an AOFS. The measurement experiment was set up using the light-beat method to calibrate the frequency shift. For comparison, the frequency shift was measured under the same conditions using a well-calibrated FPI for wind measurements as a reference system.

The new calibration system provides three advantages over the motor-driven systems using mechanical reflector disks. First of all, as a calibration instrument, higher accuracy can be acquired with the AOFS, whose characteristics indicate that its uncertainty has a negligible effect on the wind simulation. The principal uncertain source is the laser source, and the uncertainty is around 1 MHz, corresponding to a wind error of about 0.6 m/s, which is much better than the meter-level accuracy of the reflective disk. Furthermore, the new system simplifies the calibration of the experimental device, which reduces the requirements for assembly accuracy and improves operability. Finally, during calibration, the system is operated under static conditions, which improves the stability of the device and reduces safety risks.

The system has a simple structure combined with a small form factor, which can be integrated into a mobile portable device, allowing it to be used as a transfer standard to inter-calibrate a network of wind instruments.

Finally, a newly developed ASHS-type interferometer was first calibrated using the presented system, demonstrating the suitability of that measurement concept to reliably detect middle and upper atmospheric winds with a highly compact instrument.

**Author Contributions:** Y.Z.: conceptualization and methodology; G.Z.: experimentation and writing—original draft preparationresources; T.W. and W.L.: validation; M.K. and J.X.: writing—review and editing. All authors have read and agreed to the published version of the manuscript.

**Funding:** This work was supported by the Project of Stable Support for Youth Team in Basic Research Field, CAS (YSBR-018), the National Natural Science Foundation of China (41831073 and 42174196), the Chinese Meridian Project, and the Specialized Research Fund for State Key Laboratories, and the National Key R&D program of China (2021YFE0110200), the International Partnership Program of Chinese Academy of Sciences. Grant No. 183311KYSB20200003.

**Data Availability Statement:** The experimental data can be found here: http://doi.org/10.57760/sciencedb.o00009.00370 (accessed on 15 January 2023).

**Acknowledgments:** We thank the Chinese Meridian Project for the support fund. We thank the State Key Laboratory of Space Weather for the equipment in laboratory. We are also grateful to the anonymous reviewers for their constructive comments and suggestions to improve this manuscript.

**Conflicts of Interest:** The authors declare no conflict of interest.

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
