# Peer review of "An Efficient Calibration System of Optical Interferometer for Measuring Middle and Upper Atmospheric Wind"

_remotesensing, doi:10.3390/rs15071898_

Round 1

Reviewer 1 Report

The authors present a method for simulating Doppler shifted winds which can be used to calibrate various passive sensing interferometers. The method uses an acousto-optic frequency shifter instead of a spinning disc to simulate emissions from atmosphere in motion. They first calibrate the system using a photodetector and an oscilloscope then use it to calibrate an FPI and an ASHS system. The experimental work looks sound and is well presented.

A fairly well written and understandable work. As someone who works with these systems regularly it was nice to see a survey of the AOFS at several different simulated wind speeds.  

The work claims to be novel and while it is certainly recent and relevant, this method of calibration has been referenced before in at least one published work, https://doi.org/10.1016/j.jastp.2020.105363. However, a dedicated survey that explores the calibration space of the AOFS has not been published to my knowledge.

Though the flow of certain sentences would be improved with editing by a native English speaker, this is entirely optional as the work is understandable as it is. 

Some specific edits I'd recommend:

Line 67    "that" should be "which"
Line 202  has a spare period after "sensitivity"
Line 209  "speed" should be plural, or better yet, "average value of nine Doppler speed retrievals." 
Line 222  Extra space in "Doppler-shift"

I was pleased to see a more precise determination of the ASHS coefficient using statistics from 9 different tests to explain the systematic error in Figure 8.

Reviewer 2 Report

Review comments for “an efficient calibration system of optical interferometer for measuring middle and upper atmospheric wind” by Zhu et al.

This paper introduced a novel calibration system for airglow interferometers, which reduces the measurement significantly compared to using a traditional motor-driven calibration system, the latter of which is limited by the mechanical rotation speed. This system is tested on a state-of-the-art Asymmetric Spatial Heterodyne Spectrometer (ASHS) type interferometer. The instrument sensitivity is measured and corrected using this AOFS system.

This paper is neatly written. The experiments are well crafted and described. I don’t have particular concerns with the idea and results. As a reviewer with science background but less familiar with engineering, I’ve seen similar spaceborne interferometer that had been carrying onboard laser calibration system that sounds similar to this concept (e.g., WINDII on UARS as mentioned). So I have some concerns of the novelty of this current work. Other than that, there are only small issues here and there that I think can be improved. I recommend minor correction with adding in a thorough literature review and emphasis of the novelty of your work (see my major comment #1).

Major concerns:

(1) Literature review is lacking for AOFS application to instrument calibration/testing. AOFS is some quite mature technology. Similar technology has been used many years ago for onboard calibration for wind interferometers (e.g., Gerald et al., doi: 10.1117/12.56094; Sheperd et al., doi: 10.1029/2012RG000390). It is necessary to elaborate in the introduction or discussion section the novelty and advancement of your current system. In addition, how is your calibration system compared with the performance/cost/size from state-of-the-art calibration systems? If your system outperforms every other state-of-the-art instrument but cost significantly more, it is not very valuable, right?

(2) As noted by Eqn. 1 and context, the precision (i.e., sensitivity) is thermal sensitive, but it was not mentioned in the last experiment how thermal temperature is adjusted, and how the measured sensitivity responds to the temperature change. It’s critical because it’s mentioned in Lin 224 that it’s intended to be running for a long time in field stations.  

(3) integration time is not discussed. With the AOFS as the new calibration system, can the integration time for airglow interferometer be achieved to yield enough signal to noise ratio (SNR) that is better or at least comparable to with the motor-driven calibration system? In other words, the current presentation lacks some comparison study with controlled experiment (either using a traditional calibration system) or some sensitivity experiment.

Minor comments:

L45: A sentence is needed here to relate to your work here. Do most of the ground-based interferometers calibrated using the motor-driven generators?

L204: negative -> negatively.
